# Roscovitine enhances the bactericidal activity of the airway surface liquid of the cystic fibrosis bronchial epithelium but does not protect against *Pseudomonas aeruginosa* infection

Adrien Maupas[1], Anaëlle Muggeo[2], Pierre Vermeulen[1], Sophie Moussalih[1], Edouard Sage[3,4], Emilie Luczka-Majérus[1], Christelle Coraux[1], Thomas Guillard [2]*

**1** Université de Reims Champagne-Ardenne, INSERM, P3Cell, U 1250, Reims, France, **2** Université de Reims Champagne-Ardenne, INSERM, CHU de Reims, Laboratoire de Bactériologie-Virologie-Hygiène Hospitalière, P3Cell, U 1250, Reims, France, **3** Université Paris-Saclay, INRAE, UVSQ, VIM, Jouy-en-Josas, France, **4** Department of Thoracic Surgery and Lung Transplantation, Foch Hospital, Suresnes, France

* tguillard@chu-reims.fr

## Abstract

Cystic fibrosis (CF) is the most common genetic diseases in the Caucasian population. CFTR defects, the most common being F508del, lead to abnormal mucus accumulation. Respiratory failure caused by the resulting chronic infections is the leading cause of death in people with cystic fibrosis (pwCF). *Pseudomonas aeruginosa* is a major pathogen in CF and is responsible for a deterioration of respiratory function in pwCF. The increase of antibiotic-resistant *P. aeruginosa* strains encourages the search for alternative therapeutics for treating *P. aeruginosa* infection. *In vitro* studies have shown an interest in (R)-roscovitine (roscovitine) in the fight against bacterial infection in pwCF. Here we show a nuanced effect of roscovitine on ASL bactericidal activity and CF bronchial epithelium protection against *P. aeruginosa*. Using a 3D model of fully differentiated and functional F508del-CFTR human bronchial epithelium, we evidenced (i) an enhancement of the bactericidal activity of the airway surface liquid for 25 $\mu$M roscovitine but (ii) no limitation of the dynamic of the epithelium destruction upon roscovitine treatment whatever the concentrations. Our findings shed light on reasons for the lack of beneficial effects to prevent *P. aeruginosa* infection in pwCF treated with roscovitine in the ROSCO-CF clinical trial. We anticipate that our findings would have significant therapeutic implications in seeking to optimize roscovitine analogs.

## Introduction

People with cystic fibrosis (pwCF) are repeatedly and chronically infected with *Pseudomonas aeruginosa*, which plays a major role in the decline of their pulmonary

**Data availability statement:** All the raw data have been uploaded as Supporting information (5 supplemental excel tabs).

**Funding:** The authors thank the American Committee (American Memorial Hospital of Reims) for supporting this work. This work was funded by a grant from University of Reims Champagne-Ardenne.

**Competing interests:** The authors have declared that no competing interests exist.

function and the unfavourable course of the disease [1–5]. The treatments against *P. aeruginosa* are iterative, expensive, not without side effects and promote the emergence of multi-drug resistant strains [6]. In this context, the development of new anti-*P. aeruginosa* therapeutical agents is essential [7]. (R)-roscovitine (roscovitine) is a cyclin-dependent kinase inhibitor, which is widely studied in oncological research for its cellular anti-proliferative properties. Roscovitine has been also demonstrated to readdress the mutated cystic fibrosis transmembrane conductance regulator (CFTR) protein to the epithelial cell membrane [8], enhance the bactericidal activity of macrophages [9,10] and promote the inflammation resolution [9]. Despite such promising *in vitro* and *ex vivo* properties, it has been recently reported that roscovitine worsened infections of *Mycobacterium abscessus*, which is another pathogen in CF [11]. Roscovitine has been, however, proposed as a new option to treat pwCF and underwent a phase IIa clinical trial (ROSCO-CF), but no beneficial effect for a better protection against *P. aeruginosa* was evidenced [12].

The airway epithelium is the first line of defence against inhaled pathogens such as *P. aeruginosa* [13] thanks, in part, to antimicrobial peptides (AMP) secreted in the airway surface liquid (ASL) [14]. Restoring CFTR with roscovitine treatment should resolve the reduction of ASL acidification in pwCF and therefore restore AMP activity [15]. To better understand whether this lack of beneficial effect reported in ROSCO-CF may be due to the ineffectiveness of roscovitine in restoring the epithelium's antibacterial defence, we investigated if roscovitine could enhance the bacterial activity of ASL from pwCF epithelia and protect the CF bronchial epithelium from *P. aeruginosa* infection. We anticipate that our findings may have significant implications in repurposing the anticancer drug roscovitine to treat pwCF.

## Materials and methods

### Roscovitine

(R)-roscovitine (roscovitine) was provided by Perha Pharmaceuticals (formerly ManRos Therapeutics) and then purchased to Abcam® (Cambridge, UK) due to stock limitation. To confirm that the biological properties of commercial roscovitine were similar to those from Perha Pharmaceuticals, we confirmed the anti-proliferative activity of roscovitine on A549 cells (see below and S1 Fig in S1 File). Roscovitine was diluted in dimethyl sulfoxide (DMSO; Sigma).

### Bacterial strain and growth conditions

The commonly used laboratory strain *P. aeruginosa* PA14 UCBPP was a gift from G.B. Pier. Bacterial cultures were routinely incubated at 37°C on Brain Heart Infusion (BHI) medium (Oxoïd®) under agitation at 250 rpm or blood agar plate (bioMérieux, France). For bacterial infection experiments, PA14 was grown in BHI broth medium at 37°C with shaking (250 rpm) overnight and subcultured to mid-exponential growth phase ($OD_{600nm} = 0.8–1.2$), washed with PBS buffer (DPBS, Gibco, Invitrogen, Carlsbad, CA), and then suspended in DMEM/F-12 + GlutaMax medium without antibiotics. The $OD_{600nm}$ was adjusted precisely to 1 (+/- 0.02) for an inoculum of $10^9$ CFU/mL.

## Cell lines and culture

A549 cells were cultured as previously described [16]. Briefly, cells were cultured in Dulbecco's modified Eagle's medium (DMEM, Gibco) containing 10% foetal calf serum (FCS; Gibco).

## Human bronchial epithelial (HBE) cell culture

Human bronchial epithelial (HBE) cells were obtained from lung explants of pwCF homozygous for the F508del mutation (n = 6; median age = 28.5 years [range 20–36]; 4 males/ 2 females) kindly provided by Foch Hospital (Suresnes, France) between 10/09/2019 and 31/10/2022. The protocol was authorized by the French Ministry of Research (DC-2012–1583) with the written consent of the patients and the approval n°21–775 of the Institutional Review Board 00003888 Inserm. HBE cells were cultured at the air-liquid interface (ALI) to form well-differentiated and polarized epithelia, as previously described [17]. Briefly, bronchi were isolated from lungs and HBE cells were obtained after overnight incubation at 4°C with Pronase E (Sigma Aldrich, St Louis, MO) in RPMI 1640 medium supplemented with 20 mm HEPES (Gibco), 200 UI/mL penicillin, 200 $\mu$g/mL streptomycin, 80 $\mu$g/mL tobramycin, 100 $\mu$g/mL vancomycin, 100 $\mu$g/mL ceftazidime, 100 $\mu$g/mL meropenem, and 0.25 $\mu$g/mL amphotericin B. After amplification in CnT17 medium (CELLnTEc®, Bern, Switzerland) supplemented with antibiotics, HBE cells were detached, counted, and seeded (1,8 × 10$^5$ cells/ cm$^2$) on Transwell supports (Transwell-Clear®, 12 mm diameter polyester membranes; 0.4 $\mu$m pores; Corning, Acton, MA) coated with type IV collagen (Sigma Aldrich). HBE cells were grown in submerged conditions in CnT17 medium until confluency, then at the ALI for 5 weeks to obtain differentiated epithelia, using the bronchial epithelial cell growth medium (BEGM) in the basal compartment of the culture chamber. BEGM was composed of 1:1 DMEM/F12 (Gibco) and Bronchial Epithelial cell Basal Medium (BEBM, Lonza®, Walkersville, MD), supplemented with EGF, insulin, hydrocortisone, transferrin, epinephrine, tri-iodothyronine, Bovine Pituitary Extract (Lonza®), bovine serum albumin (BSA; Sigma Aldrich), retinoic acid (Sigma Aldrich) and with 100 UI/mL penicillin/ 100 $\mu$g/mL streptomycin (Gibco). Culture medium was changed 3 times a week and cells kept in humidified incubators at 37°C under 5% $CO_2$.

## Airway epithelial cell viability

Toxicity assays were performed using the TOX8 *in vitro* Resazurin based Toxicology assay kit (Sigma Aldrich, St Louis, MO), in CF HBE cell cultures treated for 9 days with roscovitine at 3, 6, 12, 25 and 50 $\mu$M, or with vehicle (DMSO; Sigma) as control (CTRL). Resazurin solution was diluted to 10% in BEGM and added to the apical surface of ALI cultures at Days 0, 2, 4, 7 and 9 post exposure to roscovitine. After a 2h incubation at 37°C, the solution was recovered and the $OD_{600nm}$ was measured. An increase in $OD_{600nm}$ value reflected the decrease in the number of metabolically active cells. The experiments were performed in duplicates for each HBE cell cultures (n = 3 pwCF).

## Collection of ASL

Apical secretions were collected from differentiated CF HBE cell cultures treated for 10 days with roscovitine at 3, 6, 12 and 25 $\mu$M, or with vehicle as CTRL (DMSO). After careful washes of culture apical surfaces with sterile PBS (Gibco), 50 $\mu$L of DMEM/F-12 + GlutaMAX (Gibco) were added in each well (45 $\mu$L/ cm$^2$) and the cultures were incubated for 4h at 37°C under 5% $CO_2$. The medium containing the cell secretions, we called ASL, was collected, centrifuged 10 min at 12,500 $g$ at 4°C to remove cellular debris and stored at -80°C.

## Bactericidal activity of ASL

The bactericidal activity of ASL on PA14 growth was assessed as described elsewhere [18]. The ASLs were 1:15 diluted in DMEM/F-12 + GlutaMAX (Gibco). Two microliters of PA14 (10$^9$ CFU/mL, see above) suspension were mixed with 30 $\mu$L of ASLs in 96-well microplates, incubated for 2 h 30 min at 37°C under low agitation, and then seeded on LB agar plates. As a control, PA14 was incubated with DMEM/F-12 + GlutaMAX (Gibco) (Gibco). CFU developed on Lysogeny broth (LB, BD

Difco™) agar plates were counted and ASL activity was expressed as the ratio of CFU/mL of PA14 exposed to ASL from roscovitine-treated HBE cultures over CFU/mL of PA14 exposed to ASL from untreated cultures. The experiments were performed in 5 replicates for each HBE cell cultures (n = 6 pwCF).

### Model of CF bronchial infection

Before infection, cultures were treated with roscovitine in the basolateral medium at 3, 6, 12 and 25 $\mu$M for 9 days or with vehicle (DMSO) as CTRL. The apical surfaces of CF HBE cultures were washed twice 24h before infection with sterile DPBS. Propidium iodide fluorescent dye for necrotic cells was added at 0.2% (LIVE/DEAD® BacLight Bacterial Viability Kit; ThermoFisher Scientific, Eugene, OR). PA14 suspensions ($10^9$ CFU/mL, see above) were 1:100 diluted in in DMEM/F-12 + GlutaMAX (Gibco) and 100 $\mu$L were added in the upper compartment of the ALI cell cultures. The inoculums represented $10^6$ CFU/ well (approximate multiplicity of infection (MOI) of 2) and were confirmed by plating serial dilutions on LB agar. The experiments were performed in 2 replicates for each HBE cell cultures (n = 4 pwCF).

### Time-lapse microscopy

Infected cultures were placed in the incubation chamber (37°C, 5% $CO_2$) of an inverted microscope (Axio Observer Z1, Zeiss®, Oberkochen, Germany). Using MetaMorph® software (Molecular Devices, San Jose, CA), time-lapse microscopy was performed with a CMOS ORCA Flash 4.0 V2 camera (Hamamatsu, Hamamatsu-city, Japan), to record infection evolution by image acquisitions of infected cells every 30 min for 72h (3 images/well). For cell death visualization, a phase-contrast image and a red fluorescent image were recorded in parallel at each time point. Red fluorescence signal corresponding to propidium iodide necrotic cells was captured with a 545/25-nm excitation filter, a 570-nm beamsplitter, and a 607/70 bandpass emission filter. Composites images were processed using Image J software (National Institutes of Health, USA).

### MTT assay

For A549 cell line proliferation analysis upon roscovitine treatment, cells were seeded in 12-well plates at a concentration of 30,000 cells per well (13,400 cells/ cm²). Twenty-four hours after plating, cells were treated with roscovitine at 3, 6, 12, 25 and 50 $\mu$M or with vehicle (DMSO) as CTRL. The number of living cells was evaluated using an MTT assay as described previously [16]. Briefly, A549 cells were incubated with 1 mg/mL 3-(4,5-dimethylthiazol-2-yl)-2,5-diphenyltetrazolium bromide (MTT; Sigma Aldrich) in culture medium for 2h. Cells were then lysed with propan-2-ol and the absorbance of the medium was measured at 560 nm using a microplate reader (Multiskan EX, ThermoFisher Scientific). The number of living A549 cells was evaluated in roscovitine- and vehicle-treated cultures. Number of living cells in cultures treated with roscovitine was expressed as percentage of living cells treated with vehicle. The experiments were performed in 4 replicates (n = 3).

### Statistical analysis data

Statistical analysis was performed using GraphPad Prism® version 7 software. Data were analyzed using unpaired Student t-test (viability of CF bronchial epithelial cells), using a Wilcoxon Signed Rank Test (bactericidal activity of ASL), non-parametric Friedman one-way ANOVA and a Dunn's multiple comparisons test (infection of CF bronchial epithelium), one sample t-test following a Shapiro-Wilk normality test (MTT assay for A549). P values < 0.05 were considered statistically different. Degree of significance are indicated as follows: * P < 0.05; ** P < 0.01; *** P < 0.001.

## Results

### Roscovitine is toxic to CF bronchial epithelia at 50 $\mu$M

We assessed the toxicity of roscovitine to fully differentiated and functional human F508del-CFTR bronchial epithelia at concentrations of 3, 6, 12, 25 and 50 $\mu$M with treatment for up to 9 days. We determined the toxicity by visualizing the

epithelia integrity and assessing the cell viability upon roscovitine treatment of ALI cultures from 3 pwCF. In comparison to the DMSO carrier solution, we evidenced a significant decrease of cell viability (Fig 1A, $OD_{600nm} = 0.44$ for 50 $\mu$M vs 0.37 for CTRL) concomitantly to the loss of integrity of the epithelia for 50 $\mu$M of roscovitine only and 9 days of treatment. We found that the epithelial structure was disrupted with areas of destruction and detachment (Fig 1B).

### Twenty-five $\mu$M roscovitine enhances the bactericidal activity of ASLs from CF bronchial epithelia

We tested whether roscovitine treatments could enhance the bactericidal activity of ASL against *P. aeruginosa*. The ASL of bronchial ALI cultures from 6 pwCF treated by roscovitine (3, 6, 12 and 25 $\mu$M) for 10 days, mimicking treatment to prevent bacterial infections as envisaged, were tested against PA14. The ratio of CFU/mL of PA14 exposed to ASL from roscovitine-treated epithelia over CFU/mL of PA14 exposed to ASL from untreated ALI cultures allowed us to assess the bacterial survival and consequently the bactericidal activity of ASL. With a ratio of 0.69, only 25 $\mu$M roscovitine statistically improved the ASL bactericidal activity (0.81, 0.67, 0.83 for 3, 6, and 12, respectively) (Fig 2).

### Roscovitine does not protect CF bronchial epithelia from *P. aeruginosa* infection

Despite our previous findings, we investigated whether roscovitine treatments could protect CF bronchial epithelia from *P. aeruginosa,* by slowing down epithelia destruction. Using an *in vitro* model of PA14 infection recapitulating lower respiratory tract infection due to inhaled pathogens, we monitored the infected epithelia's fate. The dynamic cell death of reconstructed CF bronchial epithelia infected with PA14 was visualized by the red luminescence of propidium iodide (Fig 3A). We determined the time of destruction of bronchial epithelia generated with bronchial cells from 4 pwCF and treated

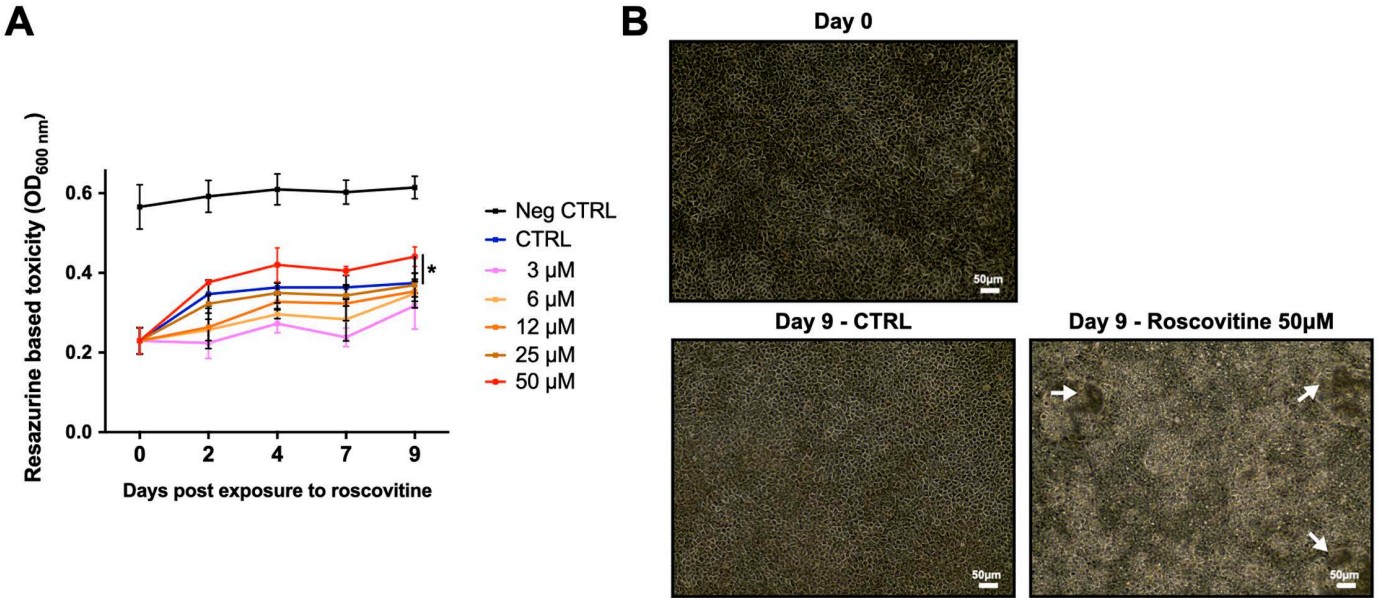

**Fig 1. Viability of CF bronchial epithelia treated with roscovitine.** (A) Resazurin based toxicity test in CF reconstructed bronchial epithelia treated with roscovitine (3, 6, 12, 25, 50 $\mu$M) for 9 days. The DMSO carrier solution was used for control 0 $\mu$M (CTRL) and the negative control (Neg CTRL) represents wells without epithelium. The Optical Density ($OD_{600nm}$) was measured 2 hours after incubation of epithelia with resazurin. Higher OD values indicate greater cell viability. Data represent mean values (2 replicates, n = 3 independent cultures from different pwCF). *$P < 0.05$ unpaired Student t-test. (B) Phase-contrast bright field micrograph of CF bronchial epithelia showing areas of destruction and detachment (arrows) after 9 days of treatment with 50 $\mu$M of roscovitine. Bars = 50 $\mu$m.

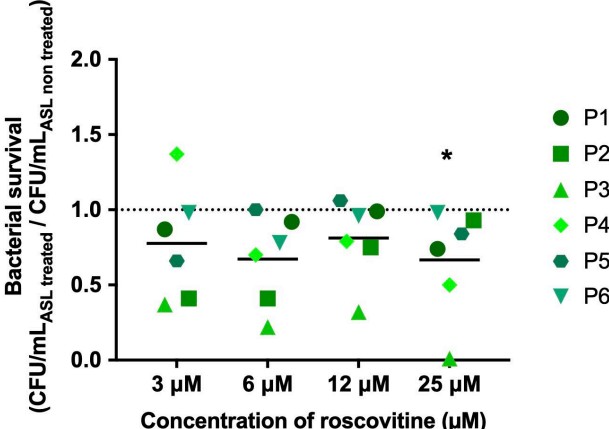

**Fig 2. Bactericidal activity against *P. aeruginosa*. of airway surface liquids from reconstructed CF bronchial epithelia treated with rosco-vitine.** The bactericidal activity of ASL from ALI cultures of CF bronchial cells from 6 independent pwCF (P1 to P6) is represented as bacterial survival (expressed as the ratio of $CFU/mL_{ASL\ roscovitine-treated}$ over $CFU/mL_{ASL\ non-treated}$). Data represent mean values of 5 replicates for each cultures. The data were analysed using a Wilcoxon Signed Rank Test.

with 3, 6, 12 and 25 $\mu$M roscovitine for 10 days before infection with PA14. We found no significant difference in epithelial destruction time, with mean values of $34 \pm 12$, $31 \pm 13$, $28 \pm 11$, $33 \pm 12$ and $31 \pm 8$h for the DMSO carrier solution (CTRL), 3, 6, 12 and 25 $\mu$M of roscovitine, respectively (Fig 3B).

## Roscovitine reagents show similar activities

Our findings were based on roscovitine purchased from two different provider. For this reason, we confirmed that commercial roscovitine was similar to roscovitine kindly provided by Perha Pharmaceuticals (formerly ManRos Therapeutics). As shown in Fig 4, we observed a decreased proliferation of A549 cells exposed to concentrations of 12, 25 and 50 $\mu$M of roscovitine either after 2 (Fig 4A) or 3 days (Fig 4B) of exposure, whatever the source of roscovitine. These concentrations

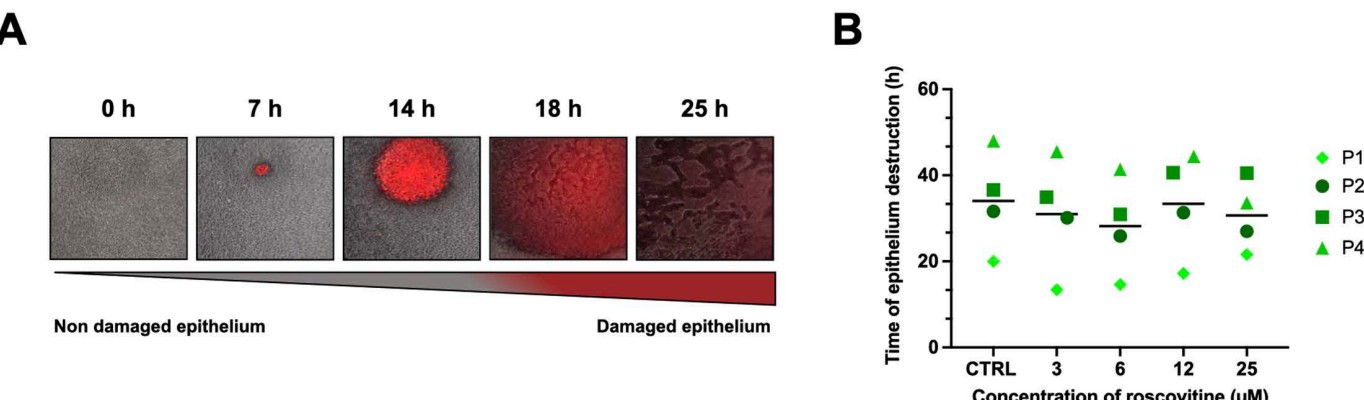

**Fig 3. Infection with *P. aeruginosa* of CF bronchial epithelia treated with roscovitine.** (A) Time lapse images of dynamic cell death of CF bronchial epithelia treated with roscovitine, then infected with PA14. In the panel, the damaged airway epithelia appear in red. (B) Time of destruction of reconstructed CF bronchial epithelia generated with cells from 4 pwCF (P1 to P4) treated with roscovitine (3, 6, 12, 25 $\mu$M) for 10 days, then infected with PA14. The DMSO carrier solution was used for control (CTRL) and represents the untreated epithelium. Data represent mean values of 2 replicates for each culture. Data were analysed using a one-way ANOVA with a P value = 0.1129 and a comparison of the mean rank of each concentration with the mean rank of the CTRL using a Dunn's multiple comparisons test.

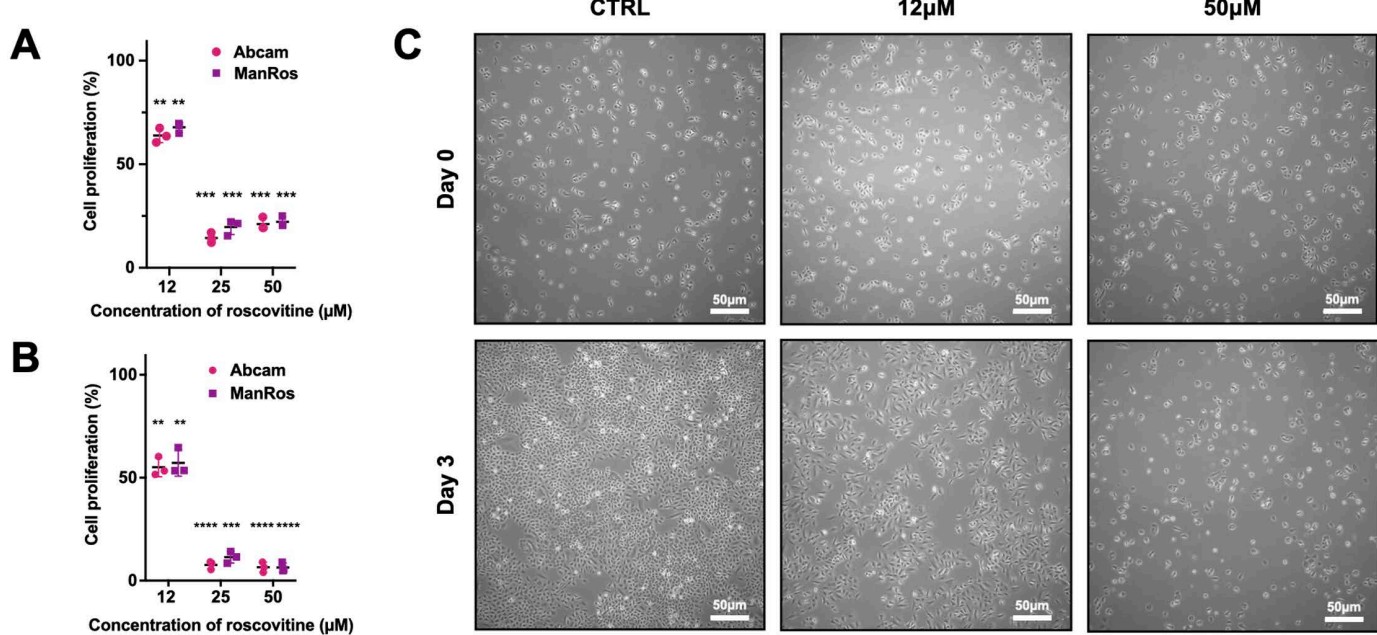

**Fig 4. Evaluation of the anti-proliferative effect of roscovitine on A549 cell line.** Cell proliferation was analysed using MTT assay after a treatment of (A) 2 or (B) 3 days with roscovitine at 12, 25 and 50 $\mu$M, purchased from Abcam or graciously gifted by Perha Therapeutics (previously ManRos Therapeutics). The data represent mean values of 4 replicates (n = 3) and error bars indicates SD. One-sample t-test, *P < 0.05; **P < 0.01; ***P < 0.001; **** P < 0.0001. (C) Phase-contrast bright field micrographs showing A549 cells before and after a treatment of 3 days with roscovitine at 12 and 50 $\mu$M. Bars = 50 $\mu$m.

were chosen according to previous reports evaluating the anti-proliferative effect of roscovitine [19]. Observations using inverted phase-contrast bright field microscope confirmed the anti-proliferative effect of roscovitine on A549 cells (Fig 4C), with less cells in roscovitine-treated cultures, compared to vehicle-treated cultures. Overall, we confirmed that commercial roscovitine was similar to roscovitine kindly provided by Perha Pharmaceuticals (formerly ManRos Therapeutics).

## Discussion

To evaluate the antiproliferative property or bactericidal activity of treated-alveolar macrophages, the studies assessing the *in vitro* effect of roscovitine used concentrations ranging from 0.1 to 100 $\mu$M [8,20]. The safety and effects of roscovitine have been evaluated in ROSCO-CF [12]. However, no data was available regarding the toxicity of roscovitine against CF bronchial epithelium at the cellular level, which precludes the assessment and justification of the concentrations used in the various studies. From our point of view, showing toxicity at 50 $\mu$M has important therapeutic implications and should be taken into account for future studies aiming at assessing the biological properties of roscovitine. First, our findings sustained that oral administration of 200 and 400 mg doses (reported as safe and well-tolerated while not 800 mg [12]) to pwCF would have no harmful impact on the bronchial epithelium. These doses are far below the minimum concentration that we tested here (3 $\mu$M ~ 1 mg/L, plasma peak concentration for 800 mg [12]). Second, it implies testing concentrations < 50 $\mu$M to evaluate the activity of roscovitine *in vitro*.

Restoring CFTR in F508del pwCF treated with roscovitine should resolve the reduction of ASL acidification in pwCF and restore AMP activity [15]. Thus, this very attractive property of roscovitine is of major concern in CF as it could prevent infection by inhaled pathogens. Our study revealed a nuanced effect of roscovitine on ASL bactericidal activity and epithelial protection in pwCF. We observed that roscovitine at a concentration of 25 $\mu$M successfully restored the bactericidal

activity of ASL. This finding is significant as it suggests that roscovitine has the potential to enhance the innate immune defenses of the CF airway in line with an improved CFTR function [15]. Despite the improvement in ASL bactericidal activity, we found that, unfortunately, roscovitine at 25 $\mu$M did not protect the CF bronchial epithelium from *P. aeruginosa* infection. These observations raised several issues on the mechanisms underlying the disconnect between improved ASL bactericidal activity and lack of epithelial protection. Further studies may help in determining other factors beyond ASL bactericidal activity that influence epithelial integrity in the presence of pathogens. Our findings questioned also if the effectiveness of roscovitine may vary depending on the infecting pathogen. It has been reported that roscovitine protected mice against pneumococcal infection [21] and Moigne *et al.* reported that roscovitine worsened *M. abscessus* infection [11]. Put in perspective of our present results, the reported antibacterial protective effect [21] could rather be the result of the anti-inflammatory property of roscovitine combined with ceftriaxone administered to mice. In addition, roscovitine for chronic treatment may have some limitations by potentially exacerbating polymicrobial infections, as *M. abscessus* and *P. aeruginosa* may both infect pwCF concomitantly. The lack of protection of the CF bronchial epithelium from *P. aeruginosa* infection may unveil, eventually, a concentration-dependent effect of roscovitine. The 25 $\mu$M concentration could be not optimal for both ASL bactericidal activity and epithelial protection. This point is crucial since we showed that roscovitine was toxic to CF bronchial epithelia at 50 $\mu$M.

## Conclusion

While our findings did not call into question the properties of roscovitine such as CFTR corrector, and enhancer of macrophage activity and inflammation resolution, they did shed light on reasons for the lack of beneficial effects to prevent *P. aeruginosa* infection in the ROSCO-CF study. Restoring the bactericidal activity of bronchial CF ASL by roscovitine appears to be more complex than initially expected for being a promising anti-*P. aeruginosa* therapeutic. Our study raises important questions that need to be addressed in future research to fully understand the potential and limitations of roscovitine in CF treatment. Overall, we anticipate that our findings would have significant therapeutic implications in seeking to optimize roscovitine analogs that can both enhance ASL bactericidal activity and protect epithelial integrity in the face of bacterial challenges.

## Supporting information

**S1 File.  Raw data of experiments showed in Figures.**
(XLSX)

## Acknowledgments

The authors thank Laurent Meijer (ManRos Therapeutics, now Perha Pharmaceuticals, Roscoff, France) for kindly providing roscovitine. The authors thank the Platform of Cell and Tissue Imaging (PICT) for imaging core facilities. The authors thank the American Committee (American Memorial Hospital of Reims) for supporting this work.

## Author contributions

**Conceptualization:** Adrien Maupas, Anaëlle Muggeo, Christelle Coraux.

**Data curation:** Adrien Maupas, Anaëlle Muggeo, Thomas Guillard.

**Formal analysis:** Adrien Maupas, Anaëlle Muggeo, Pierre Vermeulen, Sophie Moussalih, Edouard Sage, Emilie Luczka-Majérus.

**Writing – original draft:** Adrien Maupas, Thomas Guillard.

**Writing – review & editing:** Adrien Maupas, Anaëlle Muggeo, Pierre Vermeulen, Sophie Moussalih, Edouard Sage, Emilie Luczka-Majérus, Christelle Coraux, Thomas Guillard.

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
