## [Decision Letter · Decision Letter 0]

16 Dec 2024

PONE-D-24-24759Roscovitine does not protect cystic fibrosis bronchial epithelium against Pseudomonas aeruginosa infectionPLOS ONE

Dear Dr. Guillard,

Thank you for submitting your manuscript to PLOS ONE. After careful consideration, we feel that it has merit but does not fully meet PLOS ONE’s publication criteria as it currently stands. Therefore, we invite you to submit a revised version of the manuscript that addresses the points raised during the review process.

Thank you for submitting your manuscript addressing the therapeutic potential of roscovitine against *Pseudomonas aeruginosa* in cystic fibrosis (CF). This work tackles an important area of CF research, particularly given the challenges posed by multidrug-resistant pathogens. Your use of a 3D model of fully differentiated F508del-CFTR human bronchial epithelium is a notable strength, providing a robust experimental platform. However, there are certain areas, indicated below, where the manuscript can be refined to improve clarity, accuracy, and alignment with the presented data.

1) Your data presentation is generally thorough, but incorporating more direct references to numerical findings and statistical significance within the main text would enhance clarity. For instance, the observed improvement in ASL bactericidal activity at 25 µM should be quantified and discussed in greater detail. Ensuring that all conclusions are explicitly tied to the data while avoiding speculative extrapolations will improve the manuscript’s rigor and focus.

2) The manuscript would also benefit from a more explicit discussion of its limitations. The small sample size (n=6) and the in vitro nature of the experimental model limit the generalizability of the findings. Additionally, the absence of mechanistic insights into why roscovitine enhances ASL bactericidal activity at 25 µM but fails to protect epithelial integrity leaves unanswered questions. Addressing these issues in a dedicated section would provide important context for readers and help position your work within the broader research landscape.

3) The conclusions currently overstate the findings, particularly the assertion that roscovitine does not protect CF bronchial epithelium against *P. aeruginosa* . While the data clearly demonstrate the lack of epithelial protection, they also reveal enhanced bactericidal activity of the airway surface liquid (ASL) at 25 µM. This is an important observation that should be reflected more prominently. A more balanced presentation of these results, emphasizing their complexity, would strengthen the overall narrative. Revisions to the abstract, results, and discussion to reflect this duality are recommended.

4) The figures, though informative, could be made more intuitive. Figures 1 and 3, in particular, would benefit from additional annotations or more detailed captions to improve their interpretability. This would allow readers to better grasp the key findings at a glance without needing to refer back to the main text.

Overall, this manuscript has potential but requires significant revision. Specifically, the conclusions should be rephrased to reflect the nuanced findings, limitations must be discussed more thoroughly, and the presentation of data should be refined for clarity and precision. By addressing these issues, the study will make a valuable contribution to CF research and meet the standards expected for publication in *PLOS One* .

We look forward to receiving your revised manuscript.

Kind regards,

Kevin Looi, Ph.D

Academic Editor

PLOS ONE

Journal Requirements:

Reviewers' comments:

Reviewer's Responses to Questions

**Comments to the Author**

1. Is the manuscript technically sound, and do the data support the conclusions?

Reviewer #1: Yes

2. Has the statistical analysis been performed appropriately and rigorously? 

Reviewer #1: Yes

3. Have the authors made all data underlying the findings in their manuscript fully available?

Reviewer #1: Yes

4. Is the manuscript presented in an intelligible fashion and written in standard English?

Reviewer #1: Yes

5. Review Comments to the Author

Reviewer #1: The manuscript titled "Roscovitine does not protect cystic fibrosis bronchial epithelium against Pseudomonas aeruginosa infection" showcases a well-presented research on the effects of roscovitie - a cyclin-dependent kinase inhibitor, in cystic fibrosis - psedomonas aeruginosa co-infections. As people living with Cystic Fibrosis are prone to co-infections which may worsen their conditions. It is imperative to understand the metabolic function of treatments available for pwCF.

In this manuscript, the author argues that roscovitine, a treatment regimen for pwCF does not protect the bronchial epithelium of CF disease model against P. aeruginosa infection. From their data, they conclude that the medication only does not pose a significant antibacterial effects and previously reported effects could rather be the result of the anti-inflammatory property of roscovitine combined with ceftriaxone administered to mice.

From the data presented, the manuscript is technically sound and the data presented by the authors supports their conclusions. The statistical analysis was arguably performed accurately and with rigor.

I commend the authors for putting up this interesting research as it sheds more light into the important details as to intertwines between inflamatory response and antibiotic responses for systematic diseases.

6. PLOS authors have the option to publish the peer review history of their article (what does this mean? ). If published, this will include your full peer review and any attached files.

**Do you want your identity to be public for this peer review?** For information about this choice, including consent withdrawal, please see our Privacy Policy .

Reviewer #1: No

---

## [Author Response · Author response to Decision Letter 0]

21 Jan 2025

Dear Dr Looi,

Thank you for considering our work for potential publication in PLOS ONE. We are grateful to the referee for his interest in our article, "Roscovitine does not protect cystic fibrosis bronchial epithelium against Pseudomonas aeruginosa infection". We have addressed all the issues raised by the reviewer and have completed the different sections. Below you will find our point-by-point reply to the reviewer's comments, and you can see the modifications highlighted in yellow in the "Revised Manuscript with Track Changes".

Thank you for submitting your manuscript addressing the therapeutic potential of roscovitine against Pseudomonas aeruginosa in cystic fibrosis (CF). This work tackles an important area of CF research, particularly given the challenges posed by multidrug-resistant pathogens. Your use of a 3D model of fully differentiated F508del-CFTR human bronchial epithelium is a notable strength, providing a robust experimental platform. However, there are certain areas, indicated below, where the manuscript can be refined to improve clarity, accuracy, and alignment with the presented data.

1) Your data presentation is generally thorough, but incorporating more direct references to numerical findings and statistical significance within the main text would enhance clarity. For instance, the observed improvement in ASL bactericidal activity at 25 µM should be quantified and discussed in greater detail. Ensuring that all conclusions are explicitly tied to the data while avoiding speculative extrapolations will improve the manuscript’s rigor and focus.

We have highlighted the results of the ASL bactericidal activity at 25 µM as requested in the Results part (now lines 227-228). As requested, we have also discussed in greater details how our study revealed a nuanced effect of roscovitine on ASL bactericidal activity and epithelial protection in pwCF (now lines 294-313).

2) The manuscript would also benefit from a more explicit discussion of its limitations. The small sample size (n=6) and the in vitro nature of the experimental model limit the generalizability of the findings. Additionally, the absence of mechanistic insights into why roscovitine enhances ASL bactericidal activity at 25 µM but fails to protect epithelial integrity leaves unanswered questions. Addressing these issues in a dedicated section would provide important context for readers and help position your work within the broader research landscape.

We have discussed in greater details (now lines 294-313) and nuanced more our results. We mentioned unanswered questions: (i) the mechanisms underlying the disconnect between improved ASL bactericidal activity and lack of epithelial protection, (ii) whether the effectiveness of roscovitine may vary depending on the infecting pathogen and (iii) a concentration-dependent effect of roscovitine.

3) The conclusions currently overstate the findings, particularly the assertion that roscovitine does not protect CF bronchial epithelium against P. aeruginosa. While the data clearly demonstrate the lack of epithelial protection, they also reveal enhanced bactericidal activity of the airway surface liquid (ASL) at 25 µM. This is an important observation that should be reflected more prominently. A more balanced presentation of these results, emphasizing their complexity, would strengthen the overall narrative. Revisions to the abstract, results, and discussion to reflect this duality are recommended.

The conclusion has been rephrased to point out that our study revealed a nuanced effect of roscovitine on ASL bactericidal activity and epithelial protection in pwCF (now lines 320-326). The abstract has been modified also (now lines 53-54). The title has been modified in lines of our nuanced findings.

4) The figures, though informative, could be made more intuitive. Figures 1 and 3, in particular, would benefit from additional annotations or more detailed captions to improve their interpretability. This would allow readers to better grasp the key findings at a glance without needing to refer back to the main text.

Results (now lines 210-211) of Figure 1, 2 and 3 and the captions (now lines 216-220, 237-238) were more detailed.

Overall, this manuscript has potential but requires significant revision. Specifically, the conclusions should be rephrased to reflect the nuanced findings, limitations must be discussed more thoroughly, and the presentation of data should be refined for clarity and precision. By addressing these issues, the study will make a valuable contribution to CF research and meet the standards expected for publication in PLOS One.

---

## [Decision Letter · Decision Letter 1]

4 Mar 2025

PONE-D-24-24759R1Roscovitine enhances the bactericidal activity of the airway surface liquid of the cystic fibrosis bronchial epithelium but does not protect against Pseudomonas aeruginosa infectionPLOS ONE

Dear Dr. Gullard,

Thank you for submitting your manuscript to PLOS ONE. After careful consideration, we feel that it has merit but does not fully meet PLOS ONE’s publication criteria as it currently stands. Therefore, we invite you to submit a revised version of the manuscript that addresses the points raised during the review process.

Throughout the manuscript, be sure to correct the pwCF represents "people with CF" and not "patients with CF"

Scale bars on images should be made larger for reference.

In the discussion - it would be helpful to elaborate on some potential limitations of incorporating Roscovitine into a regimen for chronic infection, Specifically the authors provide evidence in the literature that Mycobacterium spp. infections get worse during treatment. As an emerging pathogen in CF, it is important to note that in some individuals, Pa and Mycobacteria may both be in the lung at the same time.

We look forward to receiving your revised manuscript.

Kind regards,

Subhra Mohapatra, MS PhD

Academic Editor

PLOS ONE

Journal Requirements:

Reviewers' comments:

Reviewer's Responses to Questions

**Comments to the Author**

1. If the authors have adequately addressed your comments raised in a previous round of review and you feel that this manuscript is now acceptable for publication, you may indicate that here to bypass the “Comments to the Author” section, enter your conflict of interest statement in the “Confidential to Editor” section, and submit your "Accept" recommendation.

Reviewer #2: All comments have been addressed

2. Is the manuscript technically sound, and do the data support the conclusions?

Reviewer #2: Yes

3. Has the statistical analysis been performed appropriately and rigorously? 

Reviewer #2: Yes

4. Have the authors made all data underlying the findings in their manuscript fully available?

Reviewer #2: Yes

5. Is the manuscript presented in an intelligible fashion and written in standard English?

Reviewer #2: Yes

6. Review Comments to the Author

Reviewer #2: Throughout the manuscript, be sure to correct the pwCF represents "people with CF" and not "patients with CF"

Scale bars on images should be made larger for reference.

In the discussion - it would be helpful to elaborate on some potential limitations of incorporating Roscovitine into a regimen for chronic infection, Specifically the authors provide evidence in the literature that Mycobacterium spp. infections get worse during treatment. As an emerging pathogen in CF, it is important to note that in some individuals, Pa and Mycobacteria may both be in the lung at the same time.

7. PLOS authors have the option to publish the peer review history of their article (what does this mean? ). If published, this will include your full peer review and any attached files.

**Do you want your identity to be public for this peer review?** For information about this choice, including consent withdrawal, please see our Privacy Policy .

Reviewer #2: No

---

## [Author Response · Author response to Decision Letter 1]

7 Mar 2025

Dear Dr Mohapatra,

Thank you for considering our work for potential publication in PLOS ONE.

Firstly, we have edited "patients with CF" for "people with CF" throughout the manuscript.

Secondly, the scale bars on images have been modified to be larger.

Thirdly, as suggested by reviewer #2, given the literature about Roscovitine and Mycobacterium abscessus, we have added a sentence mentioning some potential limitations of incorporating Roscovitine into a regimen for chronic infection (line 313-316).

---

## [Editor Report · Decision Letter 2]

16 Mar 2025

Roscovitine enhances the bactericidal activity of the airway surface liquid of the cystic fibrosis bronchial epithelium but does not protect against Pseudomonas aeruginosa infection

PONE-D-24-24759R2

Dear Dr. Guillard,

We’re pleased to inform you that your manuscript has been judged scientifically suitable for publication and will be formally accepted for publication once it meets all outstanding technical requirements.

Kind regards,

Subhra Mohapatra, MS PhD

Academic Editor

PLOS ONE
---

## [Editor Report · Acceptance letter]

PONE-D-24-24759R2

PLOS ONE

Dear Dr. Guillard,

I'm pleased to inform you that your manuscript has been deemed suitable for publication in PLOS ONE. Congratulations! Your manuscript is now being handed over to our production team.

Kind regards,

on behalf of

Dr. Subhra Mohapatra

Academic Editor

PLOS ONE